# Selective Termination of Autophagy-Dependent Cancers

**DOI:** 10.3390/cells13131096

**Published:** 2024-06-25

**Authors:** Ajit Roy, Melvin L. DePamphilis

**Affiliations:** 1National Cancer Institute, National Institutes of Health, 9000 Rockville Pike, Room 6N105, 10 Center Dr., Bethesda, MD 20892-0001, USA; ajit.roy@nih.gov; 2National Institute of Child Health and Human Development, National Institutes of Health, 9000 Rockville Pike, Room 4B413, 6 Center Dr., Bethesda, MD 20892-2790, USA

**Keywords:** PIKFYVE, PIP4K2C, PIP5K1C, endoplasmic reticulum stress, apoptosis, autophagy, cancer

## Abstract

The goal of cancer research is to identify characteristics of cancer cells that allow them to be selectively eliminated without harming the host. One such characteristic is autophagy dependence. Cancer cells survive, proliferate, and metastasize under conditions where normal cells do not. Thus, the requirement in cancer cells for more energy and macromolecular biosynthesis can evolve into a dependence on autophagy for recycling cellular components. Recent studies have revealed that autophagy, as well as different forms of cellular trafficking, is regulated by five phosphoinositides associated with eukaryotic cellular membranes and that the enzymes that synthesize them are prime targets for cancer therapy. For example, PIKFYVE inhibitors rapidly disrupt lysosome homeostasis and suppress proliferation in all cells. However, these inhibitors selectively terminate PIKFYVE-dependent cancer cells and cancer stem cells with not having adverse effect on normal cells. Here, we describe the biochemical distinctions between PIKFYVE-sensitive and -insensitive cells, categorize PIKFYVE inhibitors into four groups that differ in chemical structure, target specificity and efficacy on cancer cells and normal cells, identify the mechanisms by which they selectively terminate autophagy-dependent cancer cells, note their paradoxical effects in cancer immunotherapy, and describe their therapeutic applications against cancers.

## 1. What Is Autophagy Dependence?

Autophagy is an evolutionarily conserved ubiquitous process that recycles unnecessary or dysfunctional cellular components through a lysosome-dependent mechanism that provides the energy and materials necessary for growth, survival, and development. Since cancer cells can proliferate continuously under conditions where normal cells undergo quiescence [1], they develop a dependence on autophagy to the point of ‘addiction’ in order to survive under nutrient-limited conditions [2].

Autophagy dependence in cells are functionally detected in three ways: (1) dependence on autophagy core ATG genes, e.g., *ATG5* and *ATG7* [3], (2) high sensitivity to lysosomal activity inhibitors, e.g., chloroquine and hydroxychloroquine [4], and (3) sensitivity to PIKFYVE inhibitors that disrupt the endo-lysosomal pathway and disrupt lysosomal homeostasis [5,6,7,8,9].

Melanoma A375 cells, which are homozygous for the *BRAF^V600E^* mutation, were originally termed ‘autophagy-addicted’, because ablation of genes essential for autophagy in models of *BRAF^V600E^*-driven cancer impaired mitochondrial metabolism and increased the survival of BRAF^V600E^ tumor-bearing mice [10,11]. Consequently, melanoma A375 cells require autophagy for cell growth, proliferation, and viability even when cultured in rich medium, as evidenced by their sensitivity to lysosome inhibitors hydroxychloroquine and chloroquine [12]. Autophagy dependence is also revealed by sensitivity to established autophagy inhibitors and by high levels of the autophagosome-associated protein LC3B-II [5,13,14,15]. In advanced stages of tumorigenesis, many types of solid tumors, primary as well as metastatic, have been shown to depend on autophagy to cope with nutritional stress conditions that develop in the tumor microenvironment [16,17,18].

Different clinical trials to suppress autophagy-dependent cancers relied mostly on inhibitors of lysosomal degradation, e.g., chloroquine and hydroxychloroquine [19,20,21]. However, an acidic tumor microenvironment might inactivate these inhibitors, allowing cancer cells to escape the inhibitory effect of these molecules [22,23]. Small-molecule inhibitors of the PIKFYVE have been shown to disrupt lysosome homeostasis, which involves all three types of autophagy in mammals [macro-autophagy, micro-autophagy, and chaperone-mediated autophagy [24], henceforth simply termed autophagy], as well as endo-lysosomal trafficking pathways, suppressing many critical nutrient recovery and energy production pathways in PIKFYVE-dependent cancer cells. Remarkably, PIKFYVE inhibitors can selectively terminate autophagy-dependent cancer cells and pluripotent cancer stem cells that are PIKFYVE-dependent without having any such effect on normal cells [5,6,7,8,25,26,27,28].

## 2. What Makes Autophagy-Dependent Cancer Cells PIKFYVE-Dependent?

Whereas all cancer cells might become autophagy-dependent, all cancer cells are not PIKFYVE-dependent; some cancer cell lines are as resistant to PIKFYVE inhibitors as nonmalignant cells. Nevertheless, analyses of melanoma, B-cell non-Hodgkin lymphoma and multiple myeloma cell lines reveal that a significant fraction (40% to 75%) are, on average, 14- to 26-fold more sensitive to PIKFYVE inhibitors than non-malignant cells [6,27,29].

One way in which cancer cells might become dependent on PIKFYVE phosphoinositide kinase activity is through oncogenic mutations. For example, the *BRAF^V600E^* mutation [30] and oncogenic *KRAS* mutations [31] result in uncontrolled cell division and growth that can induce cell proliferation, migration, transformation and survival, which can lead to cancer. However, neither the *BRAF^V600E^* mutation [27] nor oncogenic *KRAS* mutations [32,33] are linked to PIKFYVE sensitivity or autophagy dependence. 

A second way in which cancer cells might become dependent on PIKFYVE activity is from limited *PIKFYVE* expression. The difference in the sensitivity of cancer cells and normal cells to PIKFYVE inhibitors ranges from 300- to 2000-fold [6,27]. However, PIKFYVE RNA levels in cancer cells and normal cells remain unchanged in response to PIKFYVE inhibition, and PIKFYVE protein levels in melanoma cell lines vary only 3-fold, with no specific correlation between the resistant and sensitive cell lines [27].

A third way is from limited autophagic flux, a measure of the degradation activity of cellular autophagy. However, the median IC_50_ values for PIKFYVE inhibitors are ~100-fold higher in resistant cells compared to sensitive cells, whereas the autophagic flux among different cell lines varies only 4-fold [27]. Thus, PIKFYVE-dependence does not result from differences in autophagic flux.

Other factors that affect the sensitivity of cells to PIKFYVE inhibitors include lysosomal genes *TFEB*, *CLCN7*, *OSTM1*, and *SNX10* [6] and p38 mitogen-activated protein kinases (p38MAPK) [7,34]. However, recent studies on cancer cell lines revealed that PIKFYE-dependence occurs when PIKFYVE becomes essential for biosynthesis of PIP2/PI(4,5)P_2_, a phosphoinositide essential for lysosome homeostasis and autophagy. 

## 3. The Role of PIKFYVE in Lysosome Homeostasis and Autophagy

The sensitivity of autophagy-dependent cancer cells to PIKFYVE inhibitors results from a requirement to synthesize the phosphoinositides required for lysosome homeostasis and autophagy (Figure 1A). Lysosome homeostasis is the balance between the fusion and fission of functional lysosomes that degrade macromolecules. Endosomes are vesicles that transport extracellular materials into the intracellular domain; late endosomes can become lysosomes. Both lysosomes and late endosomes require the phosphatidylinositol 3,5-bisphosphate [PI(3,5)P_2_] that is synthesized by PIKFYVE [35] (Figure 1B). Thus, PIKFYVE inhibition results in ‘cytoplasmic vacuolation’ (Figure 2A), because PI(3,5)P_2_ is required for lysosome fission but not for lysosome homotypic fusion [5,7,36,37]. In the absence of PIKFYVE activity, lysosomes continue to fuse (Figure 2B) but become defective in trafficking molecules into lysosomes and cathepsin maturation [5,7] (Figure 2C). Moreover, lysosomes fail to fuse with autophagosomes to form autolysosomes (Figure 2D).

Autophagy begins with formation of isolation membranes that collect malformed or damaged macromolecules and cytoplasmic organelles [46] (Figure 1B). Isolation membranes develop into double membrane vesicles termed autophagosomes that store the trash. When autophagosomes fuse with lysosomes to form autolysosomes, the trash is degraded. Formation of autolysosomes involves four phosphoinositides: PI3P, phosphatidylinositol 4-phosphate (PI4P), PI(3,5)P_2_, and phosphatidylinositol 4,5-bisphosphate [PIP2/PI(4,5)P_2_] [38,39,40,41,42,45,47]. PIP2 is a signaling lipid in lysosome homeostasis and autophagy [38,39,40] that is synthesized from phosphatidylinositol (PI) via two independent pathways (Figure 1C). The primary pathway for PIP2 biosynthesis utilizes PI4K isozymes to convert PI into PI4P which is then converted into PIP2 by PIP5K1 phosphoinositide kinase isozymes. The secondary pathway uses PIK3C phosphoinositide kinase isozymes to convert PI into PI3P that is then changed to PI(3,5)P_2_ by PIKFYVE. A phosphoinositide 3-phosphatase then converts PI(3,5)P_2_ into PI5P which is then converted into PIP2 by PIP4K2 phosphoinositide kinase isozymes [27]. Although PI(3,5)P_2_ is the major precursor for biosynthesis of PI5P via 3′-dephosphorylation [45], PIKFYVE can also convert PI directly into PI5P [47]. 

Remarkably, a lack of PIP5K1C resulted in the sensitivity of melanoma, colorectal carcinoma, and osteosarcoma cell lines to PIKFYVE inhibitors [27]. Although there are three PIP5K1 isozymes (Figure 1C), the sensitivity of these autophagy-dependent cancer cells to the PIKFYVE inhibition by WX8 correlated only with the comparative absence of PIP5K1C protein [27], whereas PIP5K1A protein was detected in 60% of the cells at equivalent levels in both WX8-sensitive and -insensitive cells and PIP5K1B was undetectable in most of the cell lines. In contrast, PIP5K1C protein was abundant in WX8-resistant cells and comparatively absent in WX8-sensitive cells. Cells that were WX8-insensitive could be converted into sensitive cells by suppressing PIP5K1C kinase activity either with a PIP5K1C-specific inhibitor, or a PIP5K1C-specific siRNA, or by ablation of the *PIP5K1C* gene. Thus, the viability of autophagy-dependent cells that depend on PIKFYVE for synthesis of PIP2 is sensitive to PIKFYVE inhibitors. 

## 4. Current Strategies for Selectively Inhibiting PIKFYVE Activity

(a)Identify small molecules that inhibit PIKFYVE activity

Based on their chemical structures and target specificity, small-molecule inhibitors of PIKFYVE fall into four groups (Figure 3, Table 1). Presumably, the characteristics of the lead compounds in groups A (WX8, vacuolin-1), B (apilimod, APY0201), C (YM201636) and D (ESK981) are predictive of the characteristics of other members of the same group. Group D consists of compounds whose chemical structures do not fit into groups A, B or C, and that do not share a chemical signature of their own. 

**Group A**—These inhibitors share a 1,3,5-triazin-2-amine core with one or more morpholine adducts. WX8, XB6 and XBA (as well as NDF and WWL in Group B) were discovered in a high-throughput screen for compounds that induce excess DNA replication selectively in cancer cells [57,58]. These five compounds rapidly induced cytoplasmic vacuolation, prevented lysosome fission, but not homotypic lysosome fusion, disrupted traffic into lysosomes and cathepsin maturation, and prevented lysosome fusion to autophagosomes [5] (Figure 2). Proteins required for lysosome fusion were required for cytoplasmic vacuolation [5,26]. The same characteristics are evident in Group B, which include NDF and WWL, and in vacuolin-1, which was identified by screening for small molecules that induced the LC3 protein associated with autophagosomes [48]. WX8 is a competitive inhibitor of ATP for binding to the PIKFYVE active site in situ as well as in vitro [27]. The same was true for its secondary target, PIP4K2C, thereby establishing WX8 target specificity.

Like WX8, vacuolin-1 also potently and reversibly inhibits fusion between autophagosomes and lysosomes in mammalian cells, thereby inducing the accumulation of autophagosomes [26,59]. In addition, vacuolin-1 was shown to block fusion between endosomes and lysosomes by activating RAB5A GTPase activity, thereby inhibiting endosomal trafficking and fusion between autophagosomes and lysosomes. Vacuolin-1 also appears to inhibit metastasis by binding to the CAPZB/CapZβ protein that blocks actin filament assembly and disassembly [60]. Vacuolin-1 suppresses the migration and invasion of from glioma cells by inhibition of lysosome exocytosis [61].

**Group B**—These inhibitors share a pyrimidine-4-amine core with a benzaldehyde hydrazone adduct and one or more morpholine adducts. Apilimod was originally identified as STA-5326 in a screen for inhibitors of interleukin-12 expression [62]. Subsequently, STA-5326 was identified as apilimod in a screen of clinical-stage drugs on mouse embryonic fibroblasts to detect inhibitors of cell proliferation [6]. As with group A inhibitors, group B chemistry is characteristic of kinase inhibitors [63]. Apilimod interacts with the asparagine (N1939) predicted to be located within the ATP-binding pocket of the catalytic kinase domain of PIKfyve [6]. A mutation at this site that prevents inhibition by apilimod also prevents inhibition by WX8 [7]. 

APY0201 was identified in a screen for inhibitors of the phosphatidylinositol 4,5-bisphosphate 3-kinase (PIK3CA/PI3K/*p110α*) [64,65] and subsequently characterized as a PIKFYVE inhibitor [29,51,52].

**Group C**—YM201636 was identified in a screen for phosphoinositide 3-kinase (PIK3CA/PI3K) inhibitors [64] and subsequently shown to selectively inhibit PIKFYVE [54]. It contains a pyrimidine core with a morpholine adduct. Notably, a PIKFYVE active site. mutant prevents its inhibition by either apilimod or WX8 with no inhibition by YM201636 [7]. Therefore, in contrast with WX8 and apilimod, YM201636 does not bind PIKFYVE in its ATP-binding pocket. 

Human ‘two-pore channels’ regulate Ca^2+^ release in endosomes and lysosomes. They are activated by PI(3,5)P_2_ and inhibited by either YM201636 or its analog PI-103 by blocking their open-state channel pore [66].

**Group D**—These molecules can inhibit PIKFYVE activity, but their chemical structures bear no similarities to those in Groups A, B or C, suggesting that they function allosterically. In fact, PIKFYVE is a 240 kD protein with multiple domains that could provide additional binding sites for inhibitors [41]. In three cases, ESK981 [9], HZX-02-059 [55], and L22 [56], the DiscoveRX KINOMEscan platform was used to demonstrate binding to PIKFYVE protein in vitro. However, this platform does not require ATP; it simply reports thermodynamic interaction affinities (https://www.eurofinsdiscovery.com/solution/kinomescan-technology (accessed on 11 June 2024)). Since none of these molecules have been shown to bind the ATP binding site of PIKFYVE, they presumably bind to PIKFYVE allosterically. 

ESK981 was discovered in a screen for inhibitors of vascular endothelial growth factor receptor (VEGFR) and Tie2 receptor tyrosine kinases. Thus, ESK981 is an angiogenesis inhibitor targeting kinases FLT1/VEGFR-1, KDR/VEGFR-2, and TEK/Tie-2 [67] that also inhibits PIKFYVE [9]. SB203580 and SB202190 are p38 mitogen-activated protein kinase (p38MAPK) inhibitors that induce cytoplasmic vacuolation by inhibiting LAMP2 phosphorylation [7] as well as PIKFYVE activity, but they are significantly weaker than YM201636 [34]. HZX-02-059 and L22 were identified by screening for compounds that induce cytoplasmic vacuolation [68], and then subsequently identified as inhibitors of protein kinases, among which was PIKFYVE [55,56,69].

(b)Identify small molecules that inhibit either VAC14 or FIG4

In vivo, PIKFYVE exists as a large heterotrimeric complex consisting of the PIKFYVE, VAC14, and FIG4 proteins [41]. PIKFYVE is a phosphoinositide kinase that can also phosphorylate itself. FIG4 has dual phosphatase activity that can dephosphorylate both the auto-phosphorylated form of PIKFYVE protein and PI(3,5)P_2_ phosphoinositide. VAC14 is a scaffolding protein [44,70]. The fact that 10 human diseases have been identified that are associated with mutations in one or more of these three genes [35,44] suggests that some PIKFYVE inhibitors might act allosterically by inhibiting either VAC14 or FIG4.

(c)Target the PIKFYVE protein for degradation

A ‘proteolysis targeting chimera’ is a heterobifunctional molecule that binds the target protein to one end and an E3 ubiquitin ligase to the other end. Thus, when introduced into cells, the target protein is sequestered and taken through the ubiquitin–proteasome system [71]. The advantage of this technique is that it depletes both the catalytic and non-catalytic functions of the target protein. Hence, it can be more effective than chemical inhibitors. This method has been applied to the PIKFYVE protein using the apilimod molecule as bait to deliver the PIKFYVE protein to the E3 ubiquitin ligase [72]. An effective PIKFYVE degrader molecule was developed termed PIK5-12d that strongly bind and degrades PIKFYVE protein (DC50 = 1.5 nM) and outperformed both apilimod and YM201636 in suppressing prostate cancer cells growth both in vitro and in vivo.

## 5. PIKFYVE Inhibitors Have Both Primary and Secondary Targets

Given the fact that none of the PIKFYVE inhibitors were identified by screening for molecules that bound directly to the PIKFYVE protein, all of them will have primary targets at low concentrations and secondary targets that become significant at higher inhibitor concentrations (Table 1). 

**Group A**—The primary and secondary targets for WX8, both in vitro and in situ, are PIKFYVE and PIP4K2C, respectively [27]. In situ, 0.05 µM WX8 competes with ATP for binding specifically to PIKFYVE, and 1 µM WX8 competes with ATP for binding to both PIKFYVE and PIP4K2C. Neither chemical inhibition nor siRNA knock-down of the tertiary targets in vitro (MTOR) or in situ (CHUK) have any significant effect on sensitive cancer cell viability. 

WX8 binds PIP4K2C with a Kd of 0.34 µM and inhibits enzyme activity with an IC_50_ of ~1 µM by competing with ATP binding [5,27]. Knock-down of PIP4K2C in PIKFYVE-dependent cells inhibited proliferation, whereas PIP4K2C knockdown together with PIKFYVE inhibition by 0.05 µM WX8 reduced cell proliferation and induced cell death [27]. In contrast, siPIP4K2C modestly increased PIKFYVE-independent HFF1 cell proliferation, reduced p62 levels, without any significant effect on cell death in the presence of 0.05 µM WX8. Thus, dual inhibition of both PIKFYVE and PIP4K2C are required to inhibit cell proliferation and induce cell death in WX8-sensitive cells. In contrast, selective inhibition of PIP4Kγ in WX8-resistant HEK293T cells increased basal level autophagy [73]. 

**Group B**—Apilimod/STA5326 [6] and its structurally close analog; NDF [5] are highly specific for PIKFYVE protein in cultured cells. Apilimod at low nanomolar concentration leads reduced ATP levels and arrest cell proliferation of PIKFYVE sensitive cells, whereas higher nanomolar concentrations (~10–100X) are required to induce cell death through non-canonical apoptosis and activation of caspases 3 and 7 [6], suggesting that elevated levels of apilimod also have secondary targets. For example, apilimod also binds to VAC14 [6] and triggers expression of inflammatory cytokines [62].

**Group C**—The primary target for YM201636 is PIKFYVE with IC_50_ values of 0.033 µM for induction of cytoplasmic vacuolation in mouse fibroblasts and neuronal cells. YM201636 selectively suppresses biosynthesis of PI5P (IC_50_ < 0.025 µM), both PI5P and PI(3,5)P_2_ (IC_50_ = 0.8 µM). YM201636 inhibits phosphoinositide kinases PIK3CA, PIK3CB and PIK3CD, the enzymes responsible for PIP3 biosynthesis [54,74,75,76]. A kinome profile of 1 μM YM201636 confirms that PIKFYVE is the primary target with PIK3CB second and PIK3CA third [74]. Thus, YM201636 at lower concentration selectively inhibits PIKFYVE, whereas at higher concentrations it inhibits multiple target required for cell growth and survival [77]. PI-103, an analog of YM201636, is also a potent multi-protein target inhibitor inhibiting proteins such as class I phosphatidylinositol 3-kinase (PIK3CA, PIK3CB in Figure 1C), mammalian target of rapamycin complex (mTOR), and DNA-dependent protein kinase (DNA-PK) [78].

**Group D**—PIKFYVE has been shown as a primary target of ESK981 but it has multiple secondary targets such as PIP5K1A and PIP5K1C [9]. Along with PIKFYVE, ESK981 also inhibits receptor tyrosine kinases implicated in angiogenesis [9]. 

Another small molecule included in this category of inhibitors is SB202190, for which the primary targets are the alpha and beta p38 mitogen-activated protein kinases (p38MAPK). Inhibition of p38MAPK induce cytoplasmic vacuolation [79]. It also inhibits PIKFYVE at concentrations 30-fold greater than YM201636 [34]. However, SB202190 has no effect on lysosome homeostasis in cells harboring phosphomimetic mutations of the p38MAPK phosphorylation sites in the lysosomal structural protein LAMP2, and SB202190 does not induce cell death unless combined with an established PIKFYVE specific inhibitor WX8 [7].

Similarly, other PIKFYVE inhibitors such as HZX-02-059 binds strongly to PIKFYVE as well as multiple secondary targets [55,69]. Synergistic cytotoxicity between apilimod and vincristine, a specific inhibitor of microtubule polymerization, suggests that tubulin is also an important secondary target. L22 appears similar to HZX-02-059 in terms of multiple secondary targets [56].

**Groups A, B and C**—Inhibitors in these groups cause TFEB (transcription factor EB) to migrate from the cytoplasm to the nucleus where it upregulates expression of genes required for autophagy and lysosomal function [39,80,81]. PIKFYVE inhibitors also upregulate expression of *TFEB* in specific cell lines from lymphoid origin [6,7,39].

## 6. The Significance of Secondary Targets 

Selective inhibition of PIKFYVE suppresses proliferation in most cells. Thus, IC_50_ values commonly reported for PIKFYVE inhibitors are for the reduction in cell proliferation. Without exception, cell death induction requires higher inhibitor concentrations, and higher inhibitor concentrations inevitably inhibit different secondary targets in addition to the primary target PIKFYVE. Often secondary targets disrupt metabolic pathways unrelated to autophagy, such as conversion of PIP2 into PIP3. 

PIP2 biosynthesis is essential for autophagy and for the biosynthesis of PIP3 by the action of PIK3C/PI3K isozymes (Figure 1C). PIP3 is concentrated on the plasma membrane where it activates an array of regulatory proteins through the AKT signaling pathway to promote cell growth, proliferation, differentiation, migration and survival [43,82]. Thus, perturbations in PIP3 levels affect cellular homeostasis in both normal and cancer cells, which accounts for the fact that PIK3C/PI3K inhibitors induced a wide variety of toxic effects during clinical trials in cancer therapy [83]. 

Although selective inhibition of PIKFYVE will suppress the biosynthesis of both PIP2 and PIP3 in PIKFYVE-dependent cancer cells, the primary pathway for biosynthesis of PIP2 (and consequently PIP3) is phosphorylation of PI4P by PIP5K1 isozymes, because PI(3,5)P_2_ is about 125-fold less abundant than PIP2 [44]. Group A inhibitors (as exemplified by WX8) will not interfere with PIP2 and PIP3 biosynthesis in normal cells, because elevated levels of WX8 inhibit only PIKFYVE and PIP4K2C. Thus, normal cells continue to synthesize PIP2 from PI4P and then PIP3 from PIP2 (Figure 4). 

In contrast, elevated levels of Group B inhibitors (as exemplified by APY0201) and Group C inhibitors (as exemplified by YM201636) will inhibit the PIK3C isozymes as well as PIKFYVE. Therefore, they will presumably inhibit PIP3 biosynthesis in normal cells. Similarly, elevated levels of ESK981 will inhibit PIKFYVE as well as the PIP5K1 isozymes required to convert PI4P into PIP2. Therefore, elevated levels of ESK981 will presumably inhibit biosynthesis of PIP2 and PIP3 in normal cells as well as in cancer cells.

**Group A**—The primary (PIKFYVE) and secondary (PIP4K2C) target for WX8 (Figure 4) and presumably other members of group A) have been established in situ as well as in vitro [5,27]. All three isozymes of the secondary target were detected in both PIKFYVE dependent and independent cells. However, the relative abundance of PIP4K2A and PIP4K2B were much higher in resistant cells. Inhibition of PIKFYVE alone suppressed proliferation of both sensitive and resistant cells, but inhibition of both PIKFYVE and PIP4K2C induced cell death selectively in sensitive cells. Transient knockdown of PIP4K2C protein with siPIP4K2C in PIKFYVE-dependent cells inhibited cell proliferation without disrupting autophagy, whereas siPIP4K2C together with PIKFYVE inhibitor WX8 induced cell death. In contrast, the same conditions had no effect on cell death in PIKFYVE-independent cells. Thus, inhibition of both PIKFYVE and PIP4K2C was required for decrease cell proliferation and cell death induction in PIKFYVE-dependent cells. The IC_50_ for cell death is ~100X greater in resistant cells than in sensitive cells. Remarkably, inhibition of PIP4Kγ in WX8-resistant HEK293T cells induce autophagy [73], presumably by stimulating biosynthesis of PIP2 via the PIK5K1C-dependent pathway. 

**Group B**—Apilimod [6] and NDF [5] are specific for PIKFYVE protein in vitro; NDF does not bind to PIP4K2C (Table 1). Apilimod exhibited antiproliferative activity in lymphoma cell lines with an IC_50_ < 0.2 µM, whereas concentrations > 0.3 µM induce cell death through non-canonical apoptosis [6], suggesting that apilimod may have secondary targets at higher concentration. One is VAC14, a component of the PIKFYVE active form. Others would be the ability to induce expression of inflammatory cytokines [62]. Secondary targets of APY0201 are PIK3CA, PIK3CB, and PIK3CD. Thus, APY0201 (and presumably other members of Group B) will inhibit the biosynthesis of PIP3 when used at elevated concentrations (Figure 4). 

**Group C**—YM201636 is specifically bound to PIKFYVE at low concentrations, but 1μM YM201636 inhibits the three catalytic subunits for PIK3CA, PIK3CB, and PIK3CD (Table 1) [54,74]. Thus, at low concentrations YM201636 selectively inhibits PI(3,5)P2 biosynthesis, whereas at higher concentrations it also inhibits the biosynthesis of PIP3 (Figure 4). 

**Group D**—ESK981, a structurally unique PIKFYVE inhibitor, has secondary targets that are PIP5K1A and PIP5K1C [9]. Thus, ESK981 inhibits the biosynthesis of both PI(3,5)P2 and PI(4,5)P2 at a higher concentration and, consequently, PIP3 biosynthesis as well (Figure 4). In addition, ESK981 inhibits kinases implicated in angiogenesis and upregulates expression of the inflammatory chemokine CXCL10 in response to Interferon gamma [9]. CXCL10 is a chemokine that promotes anti-tumor activity by promoting T-cell infiltration of tumors [84]. The combination of an immune checkpoint inhibitor and a PIKfyve inhibitor markedly increases complete tumor regression [9,85]. Similar results were obtained with apilimod. Thus, ESK981 affects multiple pathways in cell survival, including PIKFYVE inhibition.

SB202190 and SB203580 are most effective when used in combination with a PIKFYVE active sight inhibitor such as WX8 or apilimod [7].

HZX-02-059 and L22 were each selected for their ability to induce cytoplasmic vacuolation and cell death. HZX-02-059 (1 µM) inhibited 99% of the activity of at least 8 kinases, including PIKFYVE [55]. L22 secondary targets were not identified [56].

## 7. The Effects of PIKFYVE Inhibitors in Immunotherapy

Cancer immunotherapy has been successful in restricting tumors that were unresponsive to conventional treatments. However, tumors often develop mechanisms that evade the host immune response, thereby rendering immunotherapy ineffective. In an attempt to use PIKFYVE inhibitors as therapeutic molecules, several studies examined the role of PIKFYVE on innate and systemic immunity, and more recently on its role in cancer immunity. However, some reports concluded that PIKFYVE facilitates the innate immune response [86,87,88,89], whereas others concluded that PIKFYVE suppresses it [9,85,90]. The difference lies in experimental conditions; a single type of cultured immune cell cannot trigger a response from other components of the immune system, whereas animal models have a fully active immune response. 

For example, nanomolar concentrations of the PIKFYVE inhibitors in Groups A and B effectively restricted SARS-CoV-2 replication in vitro [91]. However, multiple studies suggested that apilimod blocks antiviral immune responses, and therefore the immunosuppression observed in many COVID-19 patients might be aggravated by apilimod [92]. In fact, subsequent studies revealed that PIKFYVE inhibitors worsened disease in a COVID-19 mouse model when given either prophylactically or therapeutically [91]. Trafficking of immune cells was delayed in the PIKFYVE inhibitor treated mice, resulting in an increase in neutrophils and antigen-presenting cells, but expression of interferon-stimulated genes was decreased. Thus, the effects of PIKFYVE inhibitors on coronavirus infection in vitro were misleading as to their effects on coronavirus infection in vivo.

(a)PIKFYVE suppresses the antigen presenting potential of innate immune cells.

Macrophages, neutrophils, and dendritic cells are the major components of the innate immune response that provides the first line of defense against invading pathogens. They are also phagocytes that use their plasma membrane to engulf a large particle (≥0.5 µm), giving rise to an internal compartment termed the phagosome to neutralize antigens in a lysosome-dependent manner through a process termed phagocytosis. Since phagocytosis requires functional lysosomes, it is not surprising that ablation of the *PIKFYVE* gene would impair phagocytic function, which would then impair immunity to external threats. Mice in which the *PIKFYVE* gene has been conditionally ablated in macrophages have altered alveolar macrophages. Upon exposure to house dust mite extract, mutant mice displayed severe lung inflammation and allergic asthma accompanied by infiltration of eosinophils and lymphoid cells [86]. Similarly, antigen presentation to T-lymphocytes in major histocompatibility complex (MHC) class II requires the fusion of early phagosomes with lysosomes, a process termed phagosome maturation. PIKFYVE inhibitors blocked phagosome maturation and disrupted MHC class II presentation thereby resulting in reduced activation of CD4(+) T-lymphocytes. Thus, PIKFYVE activity is needed for the processing and presentation of antigens. In addition, either apilimod or YM201636 also inhibited the fusion of dendritic cell phagosomes with lysosomes and phagosomal acidification while the production of reactive oxygen species increased [87].

Neutrophils are the first responders to infection due to their chemotactic ability. They coordinate the innate immune response in clearing pathogens and debris and reducing inflammation. Treatment of neutrophils with either apilimod or YM201636 blocked the fusion of phagosomes with lysosomes (analogous to blocking fusion of autophagosomes with lysosomes) and prevented activation of the Rac GTPases that is required for neutrophil chemotaxis and the NADPH oxidase that generates reactive oxygen species [89]. These results suggest that PIKFYVE modulates phagosome maturation through PI(3,5)P_2_-dependent activation of TRPML1, whereas chemotaxis and ROS are regulated by PI5P-dependent activation of Rac GTPases.

Type I interferon plays a key role in antiviral responses, and PIKFYVE is required for its expression in human macrophage and dendritic cells [88]. Either apilimod or a VAC14 genetic mutation that inactivates PIKFYVE activity rapidly induces expression of the stress induced transcription repressor ATF3, which then binds to the interferon promoter and blocks transcription. These results suggest that PIKFYVE controls the Toll-like receptor-mediated induction of Type I interferon.

(b)PIKFYVE inhibitors facilitate the innate immune response against cancer

The ‘immune checkpoint blockade’ is a group of inhibitors that prevent checkpoint proteins from binding with their partner proteins, thereby preventing the ‘off signal’ from being sent. This increases the efficacy of T-lymphocytes in killing cancer cells. The PIKFYVE inhibitor ESK981 induced CXCL10 chemokine expression through the interferon-γ pathway [9]. 

Genetic depletion or inhibition of PIKFYVE with either apilimod or ESK981 enhanced the antigen presentation ability of cancer cells through MHC-I surface expression leading to improved CD8+ T-lymphocyte mobilization and cancer cell killing in vitro and in vivo [85]. Moreover, PIKFYVE inhibition also enhanced the efficacy of ‘immune checkpoint blockade’ inhibitors. These results suggest that PIKFYVE inhibitors might enhance CD8+ T-lymphocyte-dependent immunotherapies by elevating the surface expression of MHC-I in cancer cells. 

Dendritic cells are the most efficient antigen-presenting cells. They take up antigens and pathogens, generate major histocompatibility complexes, migrate from the sites of antigen acquisition to secondary lymphoid organs where they physically interact with T-lymphocytes and stimulate their activity. The PIKFYVE inhibitor apilimod enhanced dendritic cell function and mobilization in tumors by selectively altering the non-canonical NF-κB pathway [90]. Either ablation of the *PIKFYVE* gene in dendritic cells or apilimod inhibition of PIKFYVE restrained tumor growth, enhanced dendritic cell-dependent T-lymphocyte immunity, and potentiated the efficacy of the immune checkpoint blockade in tumor-bearing mouse models. 

Taken together, the results described above demonstrate that the effects of PIKFYVE inhibitors on individual components of the innate immune response in vitro can be misleading as to their effects on the innate immune response against cancer in vivo. 

## 8. The Efficacy of PIKFYVE Inhibitors In Vitro

The viability of human cancer cells has been reported to be significantly more sensitive to PIKFYVE inhibitors relative to non-malignant (‘normal’) cell lines derived from the same tissues (Table 2). Viability is frequently quantified over a period of 2 to 5 days either by a loss of ATP [5,6,27], or a decrease in the number of live cells without a corresponding increase in dead cells [5,7], or the ability to form colonies [5]. The presence or absence of the *TP53* tumor suppressor gene did not alter the sensitivity of cancer cells to PIKFYVE inhibitors [7]. However, the efficacy of PIKFYVE inhibitors in suppressing viability in vitro depends on the cell line tested, the conditions under which the cells are treated, and the inhibitor used. For example, on melanoma cells, inhibitor efficacy differs by 30-fold following the pattern apilimod > WX8 > vacuolin-1 > YM202636 [5], whereas on multiple myeloma cells it differs by 13-fold following the pattern APY0201 > YM201636 > apilimod [29].

Based on changes in the levels of cellular ATP, apilimod reduced the viability of 48 different lymphoma cell lines with IC_50_ values from 0.007 µM to 6.8 µM (median 0.13 µM), whereas the IC_50_ for 12 normal cell lines ranged from 4.5 µM to 31 µM (median 15 µM). With 11 cell lines from the same type of lymphoma (Burkitt’s), the median was 0.15 µM. Thus, cell lines derived from lymphomas are 114-fold more sensitive to PIKFYVE inhibition than normal cells [6]. Using the same ATP assay, WX8 reduced the viability of seven cancer cell lines with a median IC_50_ of 0.34 µM the viability of four normal cell lines with a median IC_50_ of 21 µM, a difference of 62-fold [27], consistent with an earlier study [5]. The median for 10 melanoma cell lines with different genetic backgrounds was 2.8 µM WX8 [27]. 

Based on changes in the number of cells, WX8 reduced cell proliferation for 12 cancer cell lines with a median IC_50_ of 0.23 µM and for 8 normal cell lines with a median IC_50_ of 2.8 µM [7]. Similarly, the median IC_50_ for SB202190 on 11 cancer cell lines was 1.5 µM and on 8 normal cell lines was 23 µM [7], whereas the median IC_50_ for ESK981 on 7 prostate cell lines was 0.08 µM [9].

Subsequent studies revealed two variables that affect the IC_50_ observed with PIKFYVE inhibitors. First, the IC_50_ values is strongly affected by the seeding density of the cells. For PIKFYVE-sensitive cells, seeding densities greater than 1000–2000 cells/cm^2^ increased their IC_50_ for WX8 as much as 10-fold [27], suggesting that inhibition of autophagy is most effective when cells are proliferating rapidly. Given that cancer cells continue to proliferate under conditions where contact inhibition drives normal cells into a quiescent state (G_0_), higher seeding densities [e.g., 7420 cells/cm^2^ [7]] will increase the rate at which normal cells exit the cell cycle, thereby reducing the IC_50_ for PIKFYVE inhibitors. ATP assays, cell proliferation assays, and colony-forming assays produced equivalent results under conditions where exponential proliferation was maintained [5]. 

A second variable is inhibitor stability. Cytoplasmic vacuolation induced by PIKFYVE inhibitors is reversible [5,6,26]. With inhibitor concentrations greater than their Kd, cytoplasmic vacuolation is apparent within 30 min [5]. However, the vacuoles gradually fade away over time, revealing that the concentration of the PIKFYVE inhibitor is diminished with time either through cellular export, cellular metabolism, or serum inactivation [25]. To maintain nanomolar concentrations of a PIKFYVE inhibitor, the culture medium must be replaced with fresh inhibitor every two days [27]. 

## 9. The Efficacy of PIKFYVE Inhibitors In Vivo

The conclusion that cancer cells are significantly more sensitive to PIKFYVE inhibitors than normal cells has been confirmed in patient-derived cancer cells (ex vivo) and xenograft tumors derived from cancer cells (in vivo) (Table 3). Moreover, pretreatment of either pluripotent cancer stem cells or cancer cells with WX8 inhibited the tumor progression in mice [8,28]. In vivo, WX8 can selectively eliminate the pluripotent cancer stem cells in a teratocarcinoma, thereby converting a malignant tumor into a benign tumor [8]. Similarly, vacuolin-1 suppressed the metastasis and tumor growth of breast cancer and melanoma in mouse models [60].

PIKFYVE inhibitors also pair well with inhibitors targeted against other enzymes. For example, PIKFYVE inhibitors in groups A and B together with inhibitors whose primary targets are p38MAPK synergistically suppress colon adenocarcinoma tumor growth in mouse xenografts [7]. Inhibitors of p38MAPK can also inhibit PIKFYVE recombinant protein in vitro, but their IC_50_ is 31-times greater than the IC_50_ for YM201636 [34]. Since the IC_50_ for p38MAPK induction of cytoplasmic vacuolation in cancer cells is 31-times greater than their ability to inhibit PIKFYVE, their activity against cancer cells does not result from their ability to inhibit PIKFYVE, but from their ability to inhibit phosphorylation of the lysosomal LAMP2 protein [7]. Similarly, combining PIKFYVE inhibitor HZX-02-059 with the tubulin inhibitor vincristine results in a significant growth reduction in double-hit lymphoma tumors [55].

The maximum anti-tumor activity of ESK981 against prostate cancer was observed in immunocompetent tumor environments where activation of interferon gamma pathway by ESK981 enhanced the expression of inflammatory chemokine CXCL10 that led to functional T-cell infiltration, thereby synergizing the therapeutic response to the immune checkpoint blockade [9].

## 10. Inhibiting Cell Proliferation Precedes Inducing Cell Death

Selective inhibition of PIKFYVE activity arrests cell proliferation without inducing cell death. *PIKFYVE* is a unique, haploid sufficient, gene whose ablation results in cell cycle arrest [25]. *PIKFYVE* nullizygous mouse embryos survive until the blastocyst stage, presumably from maternally inherited PIKFYVE protein, but embryonic fibroblasts derived from Cre-induced PIKFYVE ablation develop cytoplasmic vacuolization and arrest cell division [94]. Similarly, deletion of the *Fab1/PIKFYVE* gene in yeast impairs nuclear division, resulting in aneuploid and binucleate cells [95]. Cytotoxicity requires excessive cytoplasmic vacuolation and AKT suppression [96,97], suggesting that cell death results from inhibition of both PIP2 and PIP3 biosynthesis [82].

PIKFYVE inhibition in sensitive melanoma A375 cells prevented its proliferation whereas, the cell death induction only happened when both PIKFYVE and PIP4K2C were inhibited [27]. Thus, loss of ATP and subsequent inhibition of cell proliferation occurred with an IC_50_ of 0.05 µM WX8, whereas cell death an IC_50_ of 0.68 µM WX8. Similarly, treatment with either apilimod or vincristine alone slightly induced cell death, whereas treatment with both inhibitors together increased cell death dramatically [55].

Induction of cell death by PIKFYVE inhibitors, requires higher concentrations of PIKFYVE inhibitors, but it begins 10 to 15 h after extensive cytoplasmic vacuolization occurs [8]. Thus, the onset of cell death is concomitant with autophagy disruption, an event recognized by accumulation of proteins such as LC3-II and SQSTM1/p62 associated with autophagosome [6,7,8,27]. The IC_50_ for human foreskin fibroblasts cell death, as quantified by plasma membrane permeability, annexin-V binding, and DNA loss is at least 50-times greater than PIKFYVE-dependent cells. Thus, induction of cell proliferation occurs by inhibition of PIKFYVE, whereas induction of cell death requires additional events. 

Inducing cell death in autophagy-dependent cells requires reducing the PIP2 pool. Genetic deletion of *PIKFYVE* gene in mouse embryonic fibroblasts [45] or inhibition of PIKFYVE activity by apilimod in HeLa cells [49] decreased the PI(3,5)P_2_ pool with corresponding increase in the PI3P pool, but did not reduce the PIP2 pool. Similarly, inhibiting PIP4K2C, a protein that generate PIP2 from PI5P increased the levels of PI(3,5)P_2_, PI5P, and PI3P, the different upstream phosphoinositides in HEK293T cells with non-significant effect on the levels of PI, PI4P or PIP2 [73]. 

The danger in elevated levels of PIKFYVE inhibitors resides in the inhibition of secondary targets (Section 6, Figure 4) that will affect normal cells as well as cancer cells. ESK981 will presumably inhibit biosynthesis of both PIP2 and PIP3 from PI4P, the major pathway. APY0201 (Group B) and YM201636 (Group C) will presumably inhibit PIP3 biosynthesis in normal cells. WX8 (Group A) will inhibit PIP2 biosynthesis only in PIKFYVE-sensitive cancer cells; Group A PIKFYVE inhibitors do not affect the viability of normal cells under conditions where they induce PIKYVE-dependent cell death.

## 11. PIKFYVE Inhibitors Induce PERK-Dependent Endoplasmic Reticulum Stress

PIKFYVE inhibitors disrupt lysosome homeostasis, arrest cell proliferation, disrupt autophagy, and finally induce cell death in PIKFYVE-dependent cells. RNA sequence analysis of PIKFYVE-sensitive and PIKFYVE-resistant cells suggested that IL-24 mediated ER stress response is induced selectively in PIKFYVE-dependent cells by PIKFYVE inhibitors [28]. ER-stress results due to impaired protein glycosylation, disulfide bond formation, or overexpression of secreted proteins that exceeds the folding capacity of the ER to resolve this impairment [98,99,100]. Milder ER-stress initially arrest the protein synthesis through EIF2A phosphorylation and upregulates different protein chaperones to promote the processing and refolding of proteins till the cell recover from stress. However, if the accumulation of unfolded proteins is high and ER stress could not be resolved then it results in cell death. 

Of the three ER-stress sensors that have been described in mammalian cells (i.e., protein kinase PERK, inositol-requiring enzyme IRE1a, and ATF6 transcription factor), PIKFYVE inhibition in sensitive melanoma cells and tumors triggered the PERK-dependent ER-stress response [28]. This response includes transcription factor DDIT3/CHOP/CEBPz, a member of the C/EBP transcription factor family. Since expression of *IL24* gene is controlled by transcription factors CEBPb, JUN, and FOS, induction of PERK stem of ER stress transcription factor DDIT3 upregulated IL24 expression which resulted in melanoma cell death. Ablation of the *IL24* gene in melanoma cells prevented cell death, and ectopic expression of *IL24* induced cell death in melanoma cells but not in human fibroblasts (Figure 5). The ectopic overexpression of *IL24* induces GADD (growth arrest and DNA damage-inducible) genes downstream of the PERK-dependent ER-stress pathway that induce apoptosis [101,102,103,104,105]. 

PIKFYVE inhibitor WX8 in sensitive melanoma cells and tumors induces the PERK-dependent ER-stress response with concomitant upregulation IL24 expression. This results in induction of cell death and suppression of tumor expansion. Therefore, PIKFYVE inhibitors, either alone or in combination with other targeted drugs or genetic manipulation (including ectopic IL24 protein), exhibit significant in vitro regression of growth of PIKFYVE-sensitive cancers.

## 12. PIKFYVE Inhibitors Induce Non-Canonical Apoptosis

Twelve forms of programmed cell death have been described in mammalian cells [108], but without exception, noncanonical apoptosis is cited as the cause of death induced by PIKFYVE inhibitors [6,7,8,28,55,109]. Cell death via apoptosis is recognized by DNA fragmentation, accumulation of cells containing < 2N DNA, binding of annexin-V to detect phosphatidylserine exposure in the plasma membrane, staining with either propidium iodide or trypan blue to detect plasma membrane permeability, accumulation of γH2AX to confirm double-strand DNA breaks, and cleavage of poly(ADP-ribose) polymerase (PARP) and CASP3. Apoptosis can be either p53-dependent or p53-independent. Apoptosis is referred to as noncanonical when caspase cleavage is not detected, and induction of cell death is not inhibited by caspase inhibitors such as Z-VAD-fmk. Apoptosis is the only form of programmed cell death that employs caspases 3, 6, and 7 to degrade cellular proteins indiscriminately. 

## 13. Therapeutic Potential of PIKFYVE Inhibitors against Cancers

Of the 33 patient derived cancers for which RNA levels have been quantified, 14 of them have significantly less PIP5K1C RNA than their paired normal tissues (Figure 6), suggesting that a significant fraction of cancers will respond to treatment with PIKFYVE inhibitors due to a deficiency of PIP5K1C protein. Both apilimod and ESK981 have been licensed for clinical applications, thereby demonstrating that temporary inhibition of PIKFYVE is tolerated by humans under conditions in which these drugs exhibit efficacy. Clinical trials for apilimod and ESK981 are in progress for their efficacy against non-Hodgkin lymphoma [6], prostate cancer [9,72], and renal cell carcinoma [110], as well as SARS-CoV-2 [111], and amyotrophic lateral sclerosis [45,112], Therapeutic applications for chemical inhibitors of PIKFYVE, as well as the phosphoinositide 3-kinases (PI3Ks) and phosphoinositide 4-kinases (PI4Ks) (Figure 3C), have been recently reviewed [113].

A subgroup of most, perhaps all, cancers are sensitive to chloroquine or hydroxychloroquine [115] and thus they are autophagy-dependent. Most autophagy-dependent cancers are PIKFYVE-dependent, and their dependency can be related to a intracellular deficiency in PIP5K1C protein [27]. Therefore, clinically PIKFYVE-dependent cancers could be identified by comparing the ratio of PIP5K1C to PIKFYVE protein in a cancer biopsy with a biopsy from normal tissue. However, the efficacy of PIKFYVE inhibitors against PIKFYVE-dependent cancers will also be determined by the inhibitor’s secondary targets. 

Multiple myeloma cells depend on autophagy for survival [116], and the Group B inhibitor APY0201 exhibits therapeutic potential against multiple myeloma in vivo as well as in vitro [29]. APY0201 was reported to be more effective than either apilimod or YM201636. APY0201 has also been reported effective against gastric cancer cells [52]. Given that the Group B inhibitor apilimod has been clinically licensed, it is likely that APY0201 would also clear a phase I trial. However, elevated levels of APY0201 inhibit both PIKFYVE-dependent biosynthesis of PIP2 and PIK3C-dependent biosynthesis of PIP3. Therefore, elevated levels of APY0201 (and presumably apilimod) would likely reduce the viability of normal cells. 

ESK981 can inhibit the PIP5K1 isozymes as well as PIKFYVE. Therefore, ESK981 can suppress both PIP5K1 dependent biosynthesis of PIP2 and PIP3 in normal cells, as well as inhibiting multiple tyrosine kinases. Therefore, elevated levels of ESK981 would likely reduce the viability of both normal and PIKFYVE-dependent cancer cells. On the other hand, ESK981 also augments immunotherapy in clinical settings [85], thereby facilitating its ability to terminate cancer cells. 

Of the four groups of PIKFYVE inhibitors, only those in Group A would selectively terminate PIKFYVE-dependent (autophagy-dependent) cancer cells without suppressing the viability of normal cells. Elevated levels of WX8 inhibit only PIKFYVE and PIP4K2C, thereby effectively inhibiting PIP2 biosynthesis only in PIKFYVE-dependent cells [27]. Even at elevated levels of WX8, normal cells continue to produce PIP2 and PIP3 via the major pathway of PI → PI4P → PIP2 → PIP3. 

Although PIKFYVE inhibitors can be used as a single agent against cancers, combining a PIKFYVE inhibitor with another anti-cancer therapy amplifies its therapeutic potential. Either a monoclonal antibody against CD20 cell surface antigen, or a monoclonal antibody that binds to the programmed death ligand 1 (PD-L1) amplifies the ability of apilimod to restrict the growth of Burkitt’s lymphoma [6]. The tubulin inhibitor vincristine amplifies the ability of apilimod to restrict the growth of double hit lymphoma [55]. The p38MAPK inhibitor SB202190 amplifies the ability of WX8 to restrict the growth of colorectal cancer tumors [7]. Ectopically expressed interleukin-24 amplifies the ability of WX8 to kill melanoma cells [28]. Monoclonal antibody against the programmed death ligand PD-L1 amplifies the ability of ESK981 to restrict the growth of prostate tumors [9]. 

Given the fact that PIKFYVE is essential for the activities of endosomes, lysosomes and autophagy, it is not surprising that some PIKFYVE inhibitors are more effective than others for a particular cancer. For example, APY0201 is more effective than YM201636, which is more effective than apilimod on multiple myeloma [29]. WX8 can convert malignant teratocarcinoma xenografts into benign teratomas by selectively eliminating the pluripotent cancer stem cells [8]. Applications are limited only by our understanding of the various roles played by the five phosphoinositides associated with eukaryotic cellular membranes.

## 14. Conclusions

The PubMed database currently lists 339 publications with the term ‘PIKFYVE’ in the title or abstract, thereby confirming that the PIKFYVE phosphoinositide kinase has become a popular therapeutic target. Presumably, this popularity reflects the importance of regulating the levels of phosphoinositides PI3P, PI(3,5)P_2_, PI5P, PIP2 and PIP3, because they regulate lysosome homeostasis, endosomal trafficking, and autophagy (Figure 1), as well as events that regulate cell death [117]. Thus, autophagy-dependent cancer cells that are deficient in their ability to produce PIP2 via conversion of PI4P into PIP2, must depend on PIKFYVE and PIP4K2 isozymes for biosynthesis of PIP2 and PIP3. Thus, one advantage of group A inhibitors is that they would be expected to inhibit PIP3 biosynthesis in PIKFYVE-dependent cancer cells but not in normal cells.

PIKFYVE inhibition also affects the innate immunity response and release of cytokines by virtue of the role of PIKFYVE in vesicular trafficking [118,119]. PIKFYVE inhibitors upregulate expression of inflammatory chemokines [9,120], particularly in combination with an immune checkpoint inhibitor [9,85]. PIKFYVE inhibitors can also upregulate cytokines, such as interleukin-24, which induces cell death by increasing ER-stress, particularly in combination with ectopic interleukin-24 [28]. 

Given the range metabolic effects resulting from inhibition of PIKFYVE, selectively terminating autophagy-dependent cancers is only one therapeutic target for PIKFYVE inhibitors. Clinical trials of published PIKFYVE inhibitors are underway in the treatment of SARS-CoV-2 and other RNA virus infections [121], as well as amyotrophic lateral sclerosis (ALS) [112,122] and other neurological diseases [41]. Trials are also underway for the proprietary PIKFYVE inhibitor from Verge Genomics (VRG50635) against ALS. Although apilimod was ineffective in clinical trials against Crohn’s disease and rheumatoid arthritis, the search for new PIKFYVE inhibitors, such as AS2677131 and AS2795440, to treat autoimmune and inflammatory diseases continues [120,123]. Finally, the ability of PIFYVE inhibitors to selectively eliminate pluripotent embryonal carcinoma stem cells from teratocarcinomas suggests that PIKFYVE inhibitors have therapeutic potential in treating germ cell neoplasia and in preventing spontaneous tumor formation from embryonic stem cells and induced pluripotent stem cells used in targeted gene therapy [8]. 

Based on studies published in peer reviewed journals, there are currently 15 small molecules that have been characterized as PIKFYVE inhibitors (Table 1) that can be categorized by their chemical structures into four groups (Figure 3). Groups A, B and C inhibitors compete with ATP for binding to the active site of PIKFYVE. Group D contains the PIKFYVE inhibitors that are structurally dissimilar those in groups A, B and C, and therefore assumed to bind allosterically either to the PIKFYVE protein or to the PIKFYVE-VAC14-FIG4 heterotrimer. Nanomolar concentrations of PIKFYVE inhibitors in groups A, B and C rapidly and reversibly inhibit proliferation of PIKFYVE-dependent cancer cells and pluripotent stem cells concomitant with inhibition of lysosome fission, endosomal trafficking, endosome maturation, and cathepsin maturation. Cell death results when these conditions elevate endoplasmic reticulum stress to levels that induce noncanonical apoptosis. For molecules in groups A, B and C, the IC_50_ values for inducing cytoplasmic vacuolation or inhibiting cell proliferation, for example, is significantly lower than the IC_50_ for inducing cell death. Thus, cell death results from inhibiting secondary targets such as PIP4K2C, as well as PIKFYVE. 

Group D inhibitors are not only structurally different from those in groups A, B and C, but they are dissimilar from one another. Thus, it is not surprising that secondary effects are involved as well. ESK981 can inhibit both the PIP5K1C- and PIKFYVE-dependent pathways for PIP2 synthesis, as well as activating components of the innate immune response. Thus, it has the ability to negatively affect normal cells as well as autophagy-dependent cancer cells. HZX-02-059 inhibits at least seven other kinases equally as well as it inhibits PIKFYVE. Thus, its ability to selectively terminate autophagy-dependent cells remains to be demonstrated. The primary targets for SB202190 and SB203580 are the p38MAP kinases; PIKFYVE is a secondary target. These molecules are most effective when combined with a PIKFYVE inhibitor from group A or B [7]. One should anticipate that biotech companies will produce a plethora of PIKFYVE inhibitors in the coming years, each designed to target a different disease. This story is not over; it has just begun.

## Figures and Tables

**Figure 1 cells-13-01096-f001:**
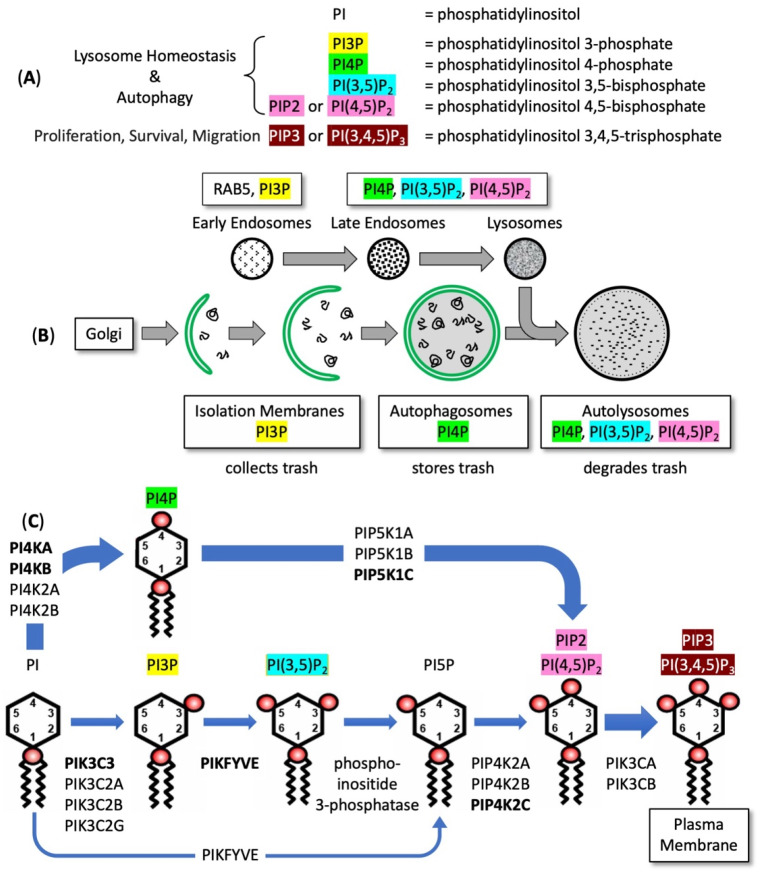
The links between phosphoinositide biosynthesis, lysosome homeostasis, and autophagy [38,39,40,41,42]. (**A**) Phosphoinositides involved with autophagy and other cellular functions. (**B**) Phosphoinositides and Rab5 associated with endosomes, lysosomes and autophagy. (**C**) Of the 22 PI kinases in the human genome [43], 15 are potentially involved in synthesizing PIP2/PI(3,5)P_2_. **Boldface** names indicate the predominant isozymes. Two additional phosphoinositide kinases are required to produce PIP3/PI(3,4,5)P_3_, a phosphoinositide that resides on the plasma membrane and activates the AKT1 kinase and related pathways that induce cell proliferation and inhibit cell death. PI(3,5)P_2_ is about 125-fold less abundant than PI(4,5)P_2_ in mouse embryonic fibroblasts [44]. Therefore, the PI4P pathway is the primary pathway for biosynthesis of PIP2 and PIP3. Similarly, the bulk of PI5P is synthesized from PI(3,5)P_2_ rather than directly from PI [45].

**Figure 2 cells-13-01096-f002:**
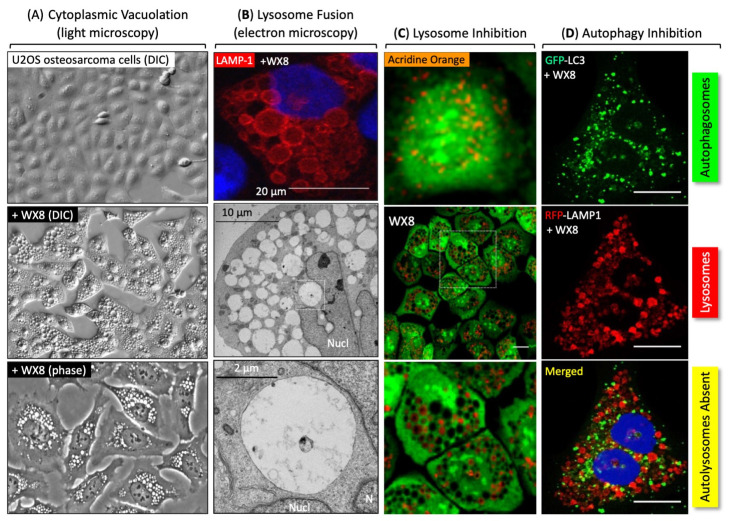
PIKFYVE inhibitors disrupt lysosome homeostasis and autophagy [5]. Using WX8 (Figure 1, Group A) as a prototype [5], (**A**) PIKFYVE inhibitors rapidly and reversibly induced cytoplasmic vacuolation, (**B**) prevented lysosome fission, but not lysosome homotypic fusion, (**C**) disrupted traffic into lysosomes and cathepsin maturation (the dotted box are magnified for better view of acridine orange exclusion), and (**D**) prevented fusion between lysosomes and autophagosomes to form autolysosomes.

**Figure 3 cells-13-01096-f003:**
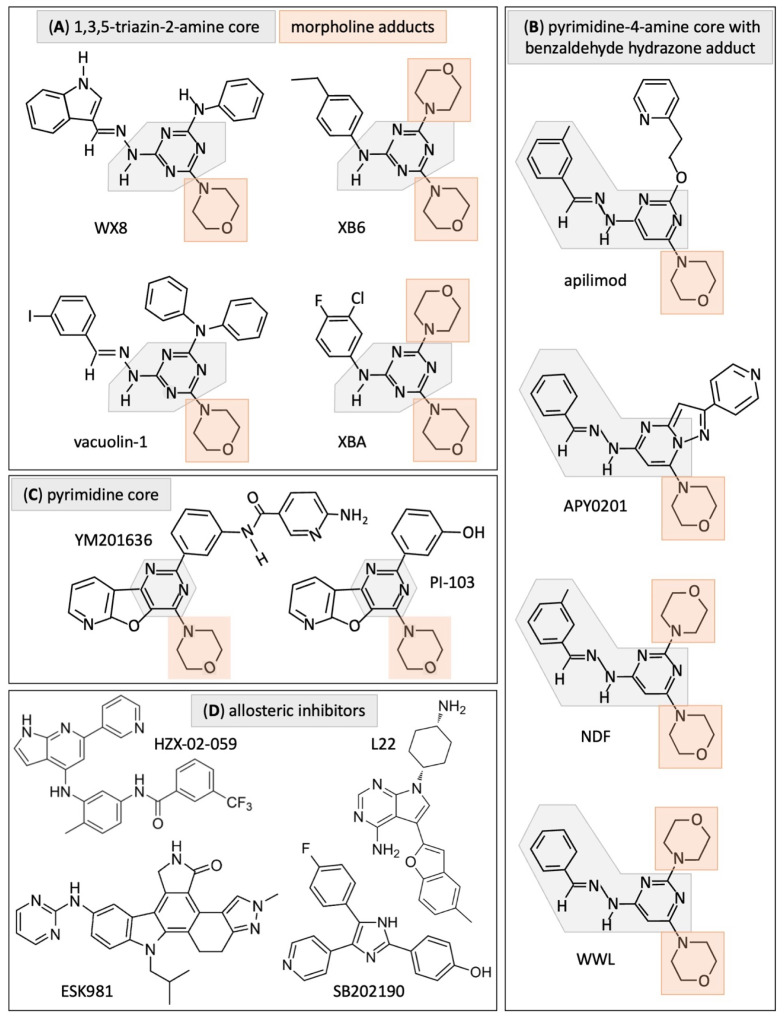
PIKFYVE inhibitors can be categorized into four groups based on chemical structures. (**A**) Compounds with a 1,3,5 triazin 2 amine core (gray) with morpholine adducts (tan). (**B**) Compounds with a pyrimidine 4 amine core with a benzaldehyde hydrazone adduct (gray) and morpholine adducts (tan). (**C**) Compounds with a pyrimidine core (gray) with a morpholine adduct (tan). (**D**) Compounds that lack homologies with groups A, B or C.

**Figure 4 cells-13-01096-f004:**
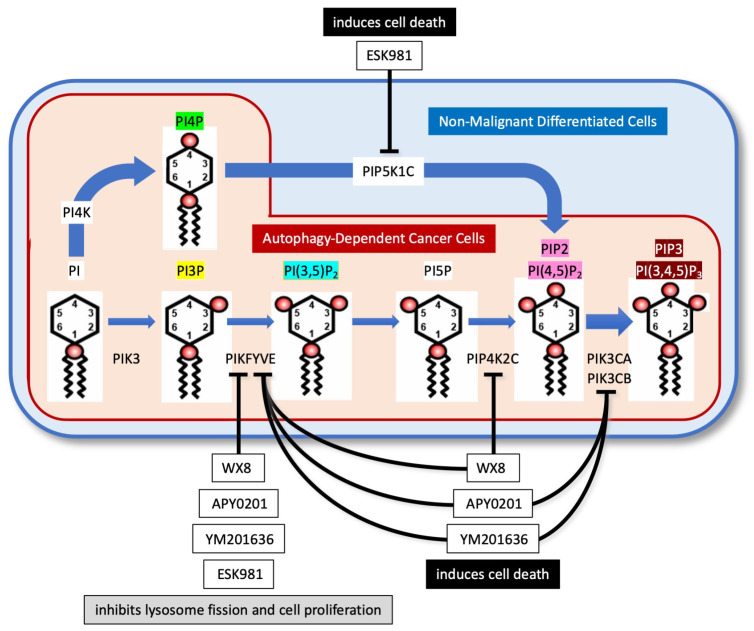
PIP5K1C phosphoinositide kinase deficiency distinguishes autophagy-dependent cancer cells from normal cells. Non-malignant differentiated cells (blue) have two pathways for PIP2/PI(4,5)P_2_ biosynthesis. One driven by PI4K and PIP5K1C and one by PIK3, PIKFYVE and PIP4K2C. Autophagy-dependent cancer cells (tan) are deficient in PIP5K1C and therefore dependent on PIK3, PIKFYVE and PIP4K2C for biosynthesis of PIP2/PI(4,5)P_2_ (Roy et al., 2023 [27]). The IC_50_ values for WX8 (Group A), APY0201 (Group B), YM201636 (Group C), or ESK981 (Group D) that selectively inhibit PIKFYVE inhibits both lysosome fission and cell proliferation. Elevated levels of PIKFYVE inhibitors that inhibit both PIKFYVE and their secondary target(s) induce cell death.

**Figure 5 cells-13-01096-f005:**
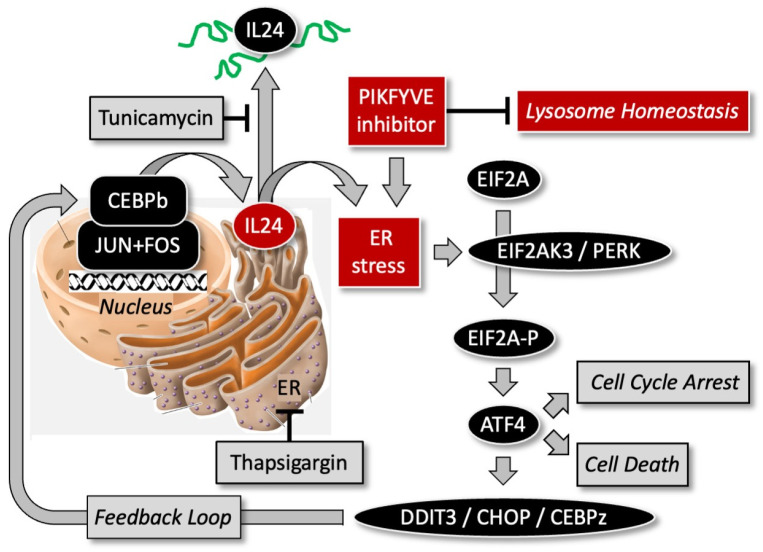
PIKFYVE inhibitors selectively kill melanoma cells by inducing interleukin-24 (IL24) expression. PIKFYVE inhibitors inhibit lysosome homeostasis, which triggers the EIF2AK3/PERK-dependent endoplasmic reticulum (ER) stress response. The PERK-dependent ER-stress response in PIKFYV-dependent melanoma cells consists of 17 genes detected by upregulation of their RNA and/or protein levels [28]. This event upregulates expression of *IL24*. Glycosylated IL24 protein is secreted, and non-glycosylated IL24 amplifies the ER-stress response to induce cell death. The same mechanism is induced by either thapsigargin or tunicamycin, but they are toxic to normal cells as well as to cancer cells. Thapsigargin induces a lethal ER-stress response by inhibiting the ATPase that regulates ER calcium homeostasis [106]. Tunicamycin induces a lethal ER-stress response by inhibiting protein glycosylation, thereby preventing proper folding and trafficking to the Golgi [107].

**Figure 6 cells-13-01096-f006:**
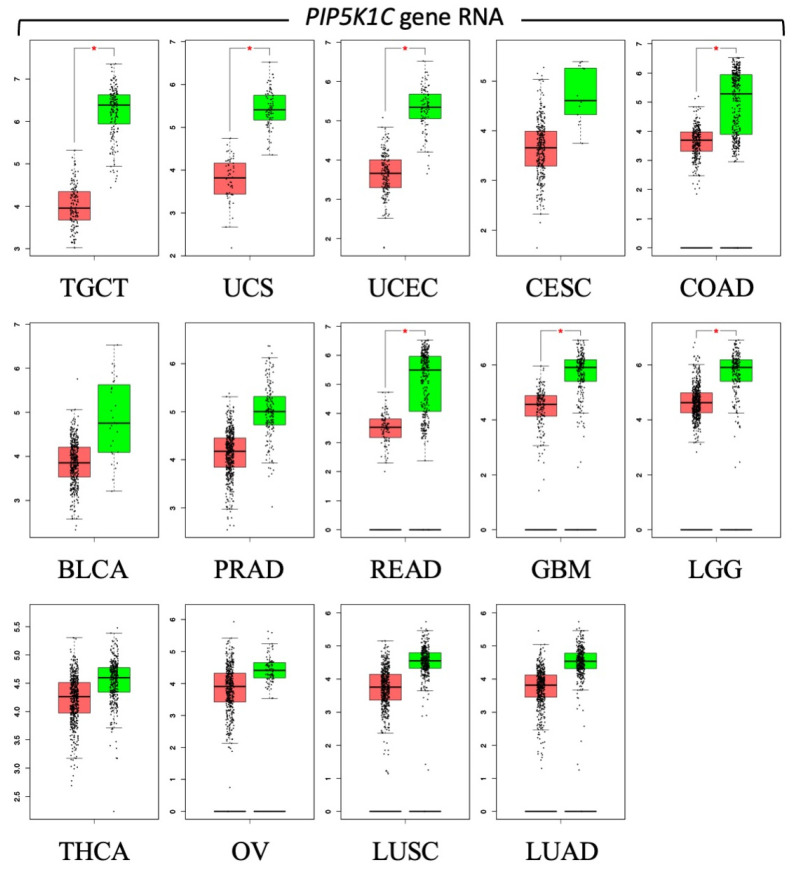
*PIP5K1C* gene expression is frequently less in cancers than in normal tissues. Box plots of 14 patient derived cancers in which the level of *PIP5K1C* gene RNA in tumor samples (red) is less than in paired normal tissues (green). Seven have *p* values ≤ 0.01 (*). The box encompasses the first quartile to the third quartile with a line through the median value. The minimum and maximum values are indicated by lines. The Y-axis is log_2_(TPM + 1) where TPM is ‘Transcripts Per Million’. Data are from the interactive web server ‘Gene Expression Profiling Interactive Analysis’ [114]. The cancers are testicular germ cell tumors (TGCT), uterine carcinosarcoma (UCS), uterine corpus endometrial carcinoma (UCEC), cervical squamous cell carcinoma and endocervical adenocarcinoma (CESC), colon adenocarcinoma (COAD), bladder urothelial carcinoma (BLCA), prostate adenocarcinoma (PRAD), rectum adenocarcinoma (READ), glioblastoma multiforme (GBM), brain lower grade glioma (LGG), thyroid carcinoma (THCA), ovarian serous cystadenocarcinoma (OV), lung squamous cell carcinoma (LUSC), and lung adenocarcinoma (LUAD).

**Table 1 cells-13-01096-t001:** Target specificity for PIKFYVE inhibitors.

Group	Inhibitor	Selection	Targets	
			Primary	Secondary
			Kd (nM)	
A	WX8	induce excess DNA replication	PIKFYVE 0.9	PIP4K2C 340
	XB6		PIKFYVE 11	PIP4K2C 990
	XBA		PIKFYVE 16	PIP4K2C 20,000
	Vacuolin-1	increase LC3 protein	PIKFYVE 39	
B	apilimod	reduce cell proliferation	PIKFYVE 0.075, 5.3, 65	VAC14
	APY0201	inhibit PIK3CA	PIKFYVE	PIK3CA PIK3CB PIK3CD
	NDF	excess DNA replication	PIKFYVE 1.6	PIP4K2C 24,000
	WWL		PIKFYVE 4.8	PIP4K2C 9200
C	YM201636	inhibit PIK3CA	PIKFYVE 9	PIK3CA PIK3CB PIK3CD
	PI-103			
D	ESK981	inhibit receptor tyrosine kinases	PIKFYVE 12	PIP5K1A 230 PIP5K1C 210
	HZX-02-059	cytoplasmic vacuolization & cell death	PIKFYVE 10 *	PIP4K2C
	L22		PIKFYVE 0.47	
	SB202190 SB203580	inhibit p38MAPK	p38MAPK	PIKFYVE

Selection criteria were applied to cultured cells. Primary target is the one with the lowest Kd. Secondary targets are the ones with the next higher Kd. Dissociation constants (Kd) were determined in vitro. * 1 µM HZX-02-059 inhibited 8 kinases 99%: PIKFYVE, DDR2, KIT, EPHA8, DDR1, KI, LCK, RAF1. PIP4K2C was inhibited 60%. **Group A** [5,27,48], **Group B** [6,29,49,50,51,52,53], **Group C** [48,54], and **Group D** [9,34,55,56].

**Table 2 cells-13-01096-t002:** PIKFYVE Inhibitors Selectively Terminate Cancer Cells In Vitro.

Inhibitor	Median IC_50_ ( µM)				Reference
	Cancer Cells		Normal Cells		
WX8	7 different cancers	0.34	4 different	21	[27]
	10 melanomas	2.8			
	12 different cancers	0.24	8 different	2.8	[7]
apilimod	48 lymphomas	0.13	12 different	15	[6]
	11 Burkitt’s lymphomas	0.15			
APY0201	5 gastric cancers	0.1–1			[52]
YM201636	2 hepatocellular carcinomas	1–5			[93]
ESK981	7 prostates	0.08			[9]
SB202190	12 different cancers	4.4	8 different	23	[7]
	recombinant PIKFYVE protein	0.4 *			[34]
	1 prostate	12.2 **			
HZX-02-059	3 lymphomas	0.19			[55]
L22	1 breast	0.23			[56]

The number and type of cells are indicated. IC_50_ is the inhibitor concentration that reduced signal by 50% under the conditions employed. * IC_50_ 0.013 µM YM201636 = 31X less than IC_50_ for SB202190. ** IC_50_ for cytoplasmic vacuolation.

**Table 3 cells-13-01096-t003:** PIKFYVE inhibitors selectively terminate cancer cells ex vivo and in vivo.

Inhibitor	Cancer	Tumor	Reference
WX8	melanoma	xenografts	[27]
	embryonal carcinoma	ex vivo xenografts	[8]
WX8 ± SB202190	colon adenocarcinoma	xenografts	[7]
vacuolin-1	melanoma breast	transgenic mouse models	[60]
apilimod	lymphoma	xenografts	[6]
APY0201	gastric	xenografts	[52]
APY0201 apilimod YM201636	multiple myeloma	ex vivo	[29]
YM201636	hepatocellular carcinoma	xenografts	[93]
ESK981	prostate	ex vivo xenografts	[9]
HZX-02-059	lymphoma	xenografts	[55]
L22	breast	xenografts	[56]

Ex vivo means the experiment was performed in patient derived cells or tissues from the indicated cancers. In vivo means the experiment was performed within the xenografts generated from cell lines derived from the indicted cancers.

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
