# Peer review of "Selective Termination of Autophagy-Dependent Cancers"

_cells, 2024, doi:10.3390/cells13131096_

Round 1

Reviewer 1 Report

Comments and Suggestions for Authors

Review article entitled “Selective Termination of Autophagy-Dependent Cancers” for the journal “Cells” by Ajit Roy and Melvin L. DePamphilis is well written and comprehensive. I do have following minor comments:

1.     Line 31: Since cancer cells never rest……needs appropriate References.

2.     Authors may elaborate a little more on autophagy-dependent cancer cells, especially their prevalence among cancer types.

3.     Line 58, authors have generalized the “endosome trafficking” with cellular homeostasis. Authors need to rectify this.

4.     In figure 3, A, authors may put PI3P as yellow along with RAB5.

5.     Line 185, Formation of ……. need to put References.

6.     Line 200, authors need to revise this sentence. PIP5K1C is only one of the three isoforms of type I PIP5K responsible for PIP2 synthesis.

7.     How prevalent is PIP5K1C deficiency among cancer types? There are reports showing increased PIP5K1C expression in cancers.

8.     Authors may need to conclude manuscript with “Future study” or “Conclusion”.

Author Response

Responses to the reviewers' comments

Reviewer #1

Review article entitled “Selective Termination of Autophagy‑Dependent Cancers” for the journal “Cells” by Ajit Roy and Melvin L. DePamphilis is well written and comprehensive. I do have following minor comments:

  1. Line 31: Since cancer cells never rest……needs appropriate References.

Response

This statement has been more accurately worded and appropriate references have been cited. page 3, lines 4-7.

  1. Authors may elaborate a little more on autophagy‑dependent cancer cells, especially their prevalence among cancer types.

Response

page 4, lines 3-7.

"Whereas all cancer cells might become autophagy dependent, all cancer cells are not PIKFYVE dependent; some cancer cell lines are as resistant to PIKFYVE inhibitors as nonmalignant cells. Nevertheless, analyses of melanoma, B-cell non-Hodgkin lymphoma and multiple myeloma cell lines reveal that a significant fraction (40% to 75%) are, on average, 14 to 26 fold more sensitive to PIKFYVE inhibitors than non-malignant cells [6,27,29]."

page 18, lines 24-27.

"Of the 33 patient derived cancers for which RNA levels have been quantified, 14 of them have significantly less PIP5K1C RNA than their paired normal tissues (Fig. 6), suggesting that a significant fraction of cancers will respond to treatment with PIKFYVE inhibitors."

  1. Line 58, authors have generalized the “endosome trafficking” with cellular homeostasis. Authors need to rectify this.

Response

The term cellular homeostasis has been deleted.

Response

  1. In figure 3, A, authors may put PI3P as yellow along with RAB5.

Response

New Figure 1 includes PI3P as yellow along with RAB5 and a reference added to the figure legend.

  1. Line 185, Formation of ……. need to put References.

Response

reference has been added. page 5, line 10. page 5, line 14.

  1. Line 200, authors need to revise this sentence. PIP5K1C is only one of the three isoforms of type I PIP5K responsible for PIP2 synthesis.

Response

We clarified and expanded this point on page 5, lines 24-33

  1. How prevalent is PIP5K1C deficiency among cancer types? There are reports showing increased PIP5K1C expression in cancers.

Response

page 18, lines 24-27

" Of the 33 patient derived cancers for which RNA levels have been quantified, 14 of them have significantly less PIP5K1C RNA than their paired normal tissues (Fig. 6), suggesting that a significant fraction of cancers will respond to treatment with PIKFYVE inhibitors due to a deficiency of PIP5K1C protein."

  1. Authors may need to conclude manuscript with “Future study” or “Conclusion”.

Response

see section 14. Conclusions on page 20.

Reviewer 2 Report

Comments and Suggestions for Authors

In this review, Roy and DePamphilis provide a comprehensive analysis of PIKfyve inhibitors and their mechanism of action in blocking autophagy-dependent cancer. The review aims to summarize known PIKfyve inhibitors in the field of cancer based on the literature. With the emerging biology of PIKfyve and drug development, it is important to review the current literature on these findings. However, several statements are incorrectly interpreted from the literature.

The description of ESK981’s mechanism of action as a PIKfyve inhibitor is not accurate. The categorization of ESK981 as an allosteric inhibitor is inappropriate, and the descriptions on lines 69 and 79 are incorrect. ESK981 has been shown to have direct binding with the PIKfyve protein (PMID: 34738088) and to inhibit PIKfyve activity. Additionally, the description on line 70 stating "a chimerical protein (PIK5-12d)" is inaccurate. PIK5-12d is a PIKfyve degrader using the proteolysis-targeting chimera (PROTAC) approach (PMID: 37605297). The description of the First-in-Class PIKfyve degrader is insufficient in this review.

While it is good to discuss the clinical development of Apilimod and ESK981, more PIKfyve inhibitors in pre-clinical and clinical development are not mentioned in this review. VRG50635 is a newly developed PIKfyve inhibitor by Verge Genomics that is in clinical development (NCT06286475, NCT06215755). PIKfyve inhibitors such as AS2677131 and NSN22769 are also not mentioned.

The biology of PIKfyve has been studied in cancer and other disease models, including SARS-CoV-2 infection, arthritis, multiple sclerosis, and amyotrophic lateral sclerosis. It is important to discuss PIKfyve inhibitors studied in various diseases.

Comments on the Quality of English Language

The English in this review is clear.

Author Response

Reviewer #2

In this review, Roy and DePamphilis provide a comprehensive analysis of PIKfyve inhibitors and their mechanism of action in blocking autophagy‑dependent cancer. The review aims to summarize known PIKfyve inhibitors in the field of cancer based on the literature. With the emerging biology of PIKfyve and drug development, it is important to review the current literature on these findings. However, several statements are incorrectly interpreted from the literature.

The description of ESK981’s mechanism of action as a PIKfyve inhibitor is not accurate. The categorization of ESK981 as an allosteric inhibitor is inappropriate, and the descriptions on lines 69 and 79 are incorrect. ESK981 has been shown to have direct binding with the PIKfyve protein (PMID: 34738088) and to inhibit PIKfyve activity.

Response

Legend to figure 3 (previously figure 1) has been corrected to state "(D) Compounds that lack homologies with groups A, B or C."

page 7, lines 14 to 22. We clarified the concept of allosteric inhibitors.  

"These molecules can inhibit PIKFYVE activity, but their chemical structures bear no similarities to those in Groups A, B or C, suggesting that they function allosterically. In fact, PIKFYVE is a 240kD protein with multiple domains that could provide additional binding sites for inhibitors [40]. In three cases, ESK981 [9], HZX 02 059 [60], and L22 [61], the DiscoveRX KINOMEscan platform was used to demonstrate binding to PIKFYVE protein in vitro. However, this platform does not require ATP; it simply reports thermodynamic interaction affinities (https://www.eurofinsdiscovery.com/solution/kinomescan-technology). Since none of these molecules have been shown to compete with ATP for binding to the active site, they presumably bind to PIKFYVE allosterically."

ESK981 clearly has multiple targets, one of which is PIKFYVE.

page7, line 23-25, "ESK981 was discovered in a screen for inhibitors of vascular endothelial growth factor receptor (VEGFR) and Tie2 receptor tyrosine kinases. Thus, ESK981 is an angiogenesis inhibitor targeting kinases FLT1/VEGFR‑1, KDR/VEGFR‑2, and TEK/Tie‑2 [59] that also inhibits PIKFYVE [9]."

page 9, line 20-21, "The primary target for the ESK981 is PIKFYVE, and its secondary targets are PIP5K1A and PIP5K1C [9]. ESK981 also inhibits receptor tyrosine kinases implicated in angiogenesis [9]."

page 12, lines 5-13, "The primary target for ESK981 is PIKFYVE, but its secondary targets are PIP5K1A and PIP5K1C [9]. Thus, at low concentrations, ESK981 inhibits PIKFYVE, whereas at higher concentrations, ESK981 inhibits both pathways for PIP2 biosynthesis and, consequently, PIP3 biosynthesis as well (Fig. 4). In addition, ESK981 inhibits kinases implicated in angiogenesis and upregulates expression of the inflammatory chemokine CXCL10 in response to Interferon gamma [9]. CXCL10 is a chemokine that promotes anti‑tumor activity by promoting T‑cell infiltration of tumors [81]. The combination of an immune checkpoint inhibitor and a PIKfyve inhibitor markedly increases complete tumor regression [9,82]. Similar results were obtained with apilimod. Thus, ESK981 affects multiple pathways in cell survival, including PIKFYVE inhibition.

page 13, lines 28-29, " The PIKFYVE inhibitor ESK981 induced CXCL10 chemokine expression through the interferon‑γ pathway [9]."

Additionally, the description on line 70 stating "a chimerical protein (PIK5‑12d)" is inaccurate. PIK5‑12d is a PIKfyve degrader using the proteolysis‑targeting chimera (PROTAC) approach (PMID: 37605297). The description of the First‑in‑Class PIKfyve degrader is insufficient in this review.

Response

We expanded this topic on page 8, section 4c.

While it is good to discuss the clinical development of Apilimod and ESK981, more PIKfyve inhibitors in pre‑clinical and clinical development are not mentioned in this review.

VRG50635 is a newly developed PIKfyve inhibitor by Verge Genomics that is in clinical development (NCT06286475, NCT06215755).

PIKfyve inhibitors such as AS2677131 and NSN22769 are also not mentioned.

Response

Our review is not about PIKFYVE; it is about "Selective Termination of Autophagy Dependent Cancers" by PIKFYVE inhibitors for the special issue of "Autophagy Contribution to Cancer Therapy Resistance".

We described the application of PIKFYVE inhibitors in treatment of coronavirus infections as an example of the unanticipated effects of the immune system on therapeutic results (page 12, lines 17-25).

We provided additional examples for apilimod and ESK981 in section 13 on Therapeutic potential of PIKFYVE inhibitors against cancers. page 18, lines 29-31.

The revised manuscript now includes applications of PIKFYVE inhibitors in the treatment of non-cancer diseases in section 14. Conclusions.  page 20, lines 24-33, page 21, lines 1-2.

Round 2

Reviewer 2 Report

Comments and Suggestions for Authors Most of my comments have been addressed. One minor change suggested here, Page 8, line 16, 'Kd=1.5nM' was wrong, in the corresponding cited paper, it should be DC50=1.48nM

Comments on the Quality of English Language

The English in this review is clear.

Author Response

Thank you for your comments, we revised it.